# Effect of N-Carbamylglutamate Supplementation on Growth Performance, Jejunal Morphology, Amino Acid Transporters, and Antioxidant Ability of Weaned Pigs

**DOI:** 10.3390/ani13203183

**Published:** 2023-10-12

**Authors:** Naizhi Hu, Pei Mao, Xiaoya Xiong, Zhuangzhuang Ma, Zhijiang Xie, Mengmeng Gao, Qiujue Wu, Wenfeng Ma

**Affiliations:** College of Animal Science and Technology, Henan University of Science and Technology, Luoyang 471000, China; hunaizhiyouxiang@163.com (N.H.); maop17@163.com (P.M.); xiaoxiao21410813@163.com (X.X.); mzz3679@163.com (Z.M.); 18348250777@163.com (Z.X.); 13383955897@163.com (M.G.); wuqiujue@163.com (Q.W.)

**Keywords:** N-carbamylglutamate, growth performance, amino acid transporters, antioxidant capacity, piglet

## Abstract

**Simple Summary:**

NCG is a structural analogue of NAG, and can activate the rate-limiting enzyme of arginine synthesis and promote the synthesis of endogenous arginine in the intestine. Recent studies have found that NCG plays an important role in improving growth performance and maintaining intestinal health. However, there are few reports on the effects of NCG on the intestinal development and antioxidant capacity of piglets. The aim of this study was to investigate the effects of dietary NCG supplementation on the growth performance, apparent nutrient digestibility, jejunal morphology, amino acid transporters, and antioxidant capacity of weanling piglets. Studies have shown that dietary NCG supplementation can improve the growth performance and intestinal antioxidant capacity of weaned piglets.

**Abstract:**

Weaning is an important period that affects the performance of piglets. However, the regulation of dietary amino acid levels is considered to be an effective way to alleviate the weaning stress of piglets. N-carbamylglutamate (NCG) plays an important role in improving the growth performance and antioxidant capacity of animals. A total of 36 weaned piglets were randomly assigned to two treatment groups, a control group (CON) and a 500 mg/kg NCG group (NCG), and the experiment lasted for 28 days. The results show that the NCG treatment group showed an increased 0–28 days average weight gain and average daily feed intake, and also increased contents of GLU and HDL, and lower SUN in serum, and an upregulation of the expression of the amino acid transporters SNAT2, EAAC1, SLC3A1, and SLC3A2 mRNA in the jejunum (*p* < 0.05), as well as an increased villus length and VH:CD ratio, and claudin-1, occludin, and ZO-1 mRNA expression in the jejunum (*p* < 0.05). The NCG treatment group showed an increased content of GSH-Px in serum and T-AOC and SOD in the jejunum, and a lower content of MDA (*p* < 0.05); and the upregulation of the mRNA expression related to antioxidant enzymes (CAT, SOD1, Gpx4, GCLC, GCLM and Nrf2, AhR, CYP1A1) in the jejunal mucosa (*p* < 0.05). In addition, compared with the control group, the NCG treatment group saw an upregulation in the mRNA expression of IL-10 and a decrease in the expression of IL-1β and IL-4 in the jejunal mucosa (*p* < 0.05). In summary, the results of this study suggest that NCG improved growth performance and jejunal morphology, improved the jejunal transport of amino acids related to the ornithine cycle, and improved the antioxidant capacity in weaned pigs.

## 1. Introduction

Weaning is a significant determinant of the production rhythm in large-scale pig farming. During the weaning process, piglets encounter abrupt alterations in diet, living conditions, and social interactions that can readily induce intestinal stress [1]. This intestinal stress has the potential to affect the intestinal morphology of piglets, leading to the diminished transmembrane transport of nutrients and impaired digestion and absorption [2,3]. The jejunum, which is a crucial component of the small intestine, is responsible for nutrient absorption. It effectively assimilates a large proportion of lipids, amino acids, and other essential nutrients [4]. Recently, amino acids have garnered recognition as pivotal components for sustaining the intestinal morphology and function of piglets, significantly contributing to physiological processes such as the intestinal mucosal metabolism [5,6]. Among these amino acids, arginine and glutamine have emerged as key players in the proliferation, differentiation, and wound healing of the piglet intestinal mucosa [7,8].

N-carbamoylglutamate (NCG), an analog of metabolically stable N-acetylglutamate (NAG), has demonstrated its capability to elevate plasma arginine levels and stimulate endogenous arginine synthesis by activating carbamoyl phosphate synthase-1 (CPS-1) and pyrroline-5 carboxylate synthase (P5CS) [9,10,11]. Given the low NAG activity in the intestines of piglets, post-weaning arginine requirements often remain inadequately fulfilled, rendering arginine an essential amino acid for piglets [12]. Research conducted on poultry has indicated that NCG can enhance the performance and egg quality as well as the antioxidant capacity of laying hens [13,14]. Studies using NCG in weaned piglets have found that NCG may promote intestinal growth and improve growth performance [6]. Based on studies conducted on NCG in monogastric animals, it is proposed that dietary supplementation with NCG reduces post-weaning stress and enhances the absorption of nutrients, as well as improving antioxidant properties within the intestinal environment. Consequently, the present study was undertaken to scrutinize the impact of NCG on performance metrics, apparent nutrient digestibility, serum biochemical parameters, jejunal morphology, and the expression of amino acid transporters. By investigating antioxidant capacity, this study aims to provide foundational insights that contribute to further inquiries into the ramifications of NCG supplementation in weanling piglets.

## 2. Materials and Methods

### 2.1. N-Carbamylglutamate

NCG (white powder, purity > 98.0%) was procured from Beijing Animal Science. & Tech. Co., Ltd., Beijing, China, and incorporated into the control diet at a dose of 500 mg/kg.

### 2.2. Animals, Diets, and Experimental Design

Thirty-six Duroc × (Landrace × Yorkshire) weaned piglets aged 24 days, with an initial average body weight (BW) of 5.69 ± 0.50 kg, were selected for this study. The 36 piglets were numbered and completely randomized to the experimental or control group. No significant difference in average body weight was maintained between the two groups. There were 6 pens per treatment, with 3 pigs per pen, and the area of the pen was 120 cm × 80 cm (length × width), surrounded by stainless steel tubes and equipped with a nipple water dispenser and a feed trough. The basal diet was formulated according to NRC (2012) guidelines [15], and the mashed diet consisted of corn and soybean meal. Table 1 presents the nutritional and chemical composition of the diet. The control group was fed the basal diet, and the experimental group was fed the basal diet supplemented with 500 mg/kg NCG. The 500 g NCG and 2.5 kg of the feed were premixed, and then this 3.0 kg premix was mixed with 997 kg of the feed to obtain a concentration of 500 mg/kg of NCG in the experimental diet. The adaptation period spanned 5 days, followed by a 28-day experimental period.

### 2.3. Growth Performance and Sample Collection

The experimental facility, pens, feed containers, and water dispensers were thoroughly cleaned and disinfected before the initiation of the study. The ambient temperature within the pig housing facility was maintained between 26 °C and 28 °C, and the relative humidity was maintained at 60–70%. All pigs were provided with ad libitum access to both feed and water. The initial, mid-term, and final weights of the weaned pigs were recorded. The daily feed intake and BW were documented on days 0, 14, and 28, allowing for the calculation of parameters such as the average daily gain (ADG), average daily feed intake (ADFI), and feed ratio (F:G ratio). The formulas for calculating the ADFI, ADG, and F/G are as follows:ADFI = total feed intake/(days of feeding × number of pigs);
ADG = (final body weight − initial body weight)/days of feeding;
F:G = ADFI/ADG.

At 11–14 days and 25–28 days after the start of the experiment, cotton swabs were used to stimulate the anal defecation of 3 piglets per pen before morning feeding. The mixed 300 g of fresh feces was collected from the pens every day, and the hair was removed evenly. To the collected fresh feces samples, 10% H_2_SO_4_ solution was added at 10% of the weight of the feces to avoid nitrogen volatilization. The samples were then stored in a refrigerator at −20 °C.

At the end of the experiment, 12 piglets were randomly selected from each pen for tissue sampling and were slaughtered humanely. Blood samples were drawn from the anterior vena cava using a vacuum blood collection before slaughter and were allowed to remain static for 30 min before centrifugation (3500× *g*, 4 °C, 15 min). The resulting serum samples were harvested and stored at −20 °C. The abdominal cavity of the piglets was opened, and the jejunum was promptly excised from the intestinal body in accordance with anatomical guidelines and then sectioned to 2 cm lengths. The isolated jejunal tissue was preserved in 4% paraformaldehyde solution for subsequent jejunal morphology assessments. To prepare the jejunal mucosal samples, the jejunal contents were washed using 0.9% ice-cold saline, and the mucosal scrapings were collected on a sterile glass slide positioned on an ice-filled tray. These scrapings were then transferred to cryopreservation tubes, rapidly frozen in liquid nitrogen, and stored at −80 °C.

### 2.4. Chemical Analyses

The fecal samples were dried in a blast-drying oven (DHG-9070, Shanghai bluepard instruments Co., Ltd., Shanghai, China) at 65 °C for 72 h. The samples were cooled with a vacuum-drying apparatus and weighed. The dried sample was crushed and sieved through a 40-mesh screen. The apparent total digestibility of dry matter (DM), crude protein (CP), ether extract (EE), acid detergent fiber (ADF), neutral detergent fiber (NDF), calcium (Ca), and phosphorus (P) was determined using acid-insoluble ash as an indigestible marker. After drying at 105 °C for 24 h, the DM content was determined. The CP content was analyzed using an automatic Kjeldahl apparatus (K1100, Hanon, Jinan, China), while the EE content was analyzed using the Soxhlet extractor (JC-ST-06, Juchuang Group Co., Ltd., Jinan, China). The samples were burned in a muffle furnace to obtain crude ash, which was further treated with hydrochloric acid to determine the acid-insoluble ash (AIA) content. The Ca and P contents were analyzed through potassium permanganate titration and colorimetry, respectively. The NDF and ADF contents in the feed and feces were assessed using an automated fiber analyzer (A2000i, Ankom Technology, Macedon, NY, USA). The apparent digestibility of nutrients was evaluated using the AIA method, and the apparent nutrient digestibility was calculated in accordance with the following formula:Nutrient apparent digestibility = 1 − (A/B) × (AIA_diet_/AIA_feces_)
where A is the nutrient level in feces, and B is the nutrient level in the diet.

### 2.5. Serum Biochemical Parameter Analysis

Serum biochemical parameters, including the glucose (GLU), total protein (TP), albumin (ALB), globulin (GLB), high-density lipoprotein (HDL), low-density lipoprotein (LDL), total cholesterol (TC), triglycerides (TGs), and serum urea nitrogen (SUN), were quantified using spectrophotometric kits, according to the manufacturer’s instructions (TBA-120FR, Toshiba Medical Systems Corporation, Tokyo, Japan). Measurements were conducted using an automatic biochemistry radiometer (Au640, Olympus, Tokyo, Japan).

### 2.6. Jejunal Morphological Analysis

Jejunal samples fixed with 4% paraformaldehyde were prepared using the paraffin-embedding method. Samples were cut 5 μm thick and placed on glass slides, then stained using hematoxylin and eosin. Jejunal sections were scanned for imaging using a digital tissue slice imaging scanner (Pannoramic MIDI, 3dhistech, Budapest, Hungary). The villus height (VH), crypt depth (CD), and intestinal wall thickness (IWT) were measured using digital slice browsing software (CaseViewer 2.4, 3dhistech, Budapest, Hungary), and the calculation of the VH:CD ratio. At least 10 complete villi and associated crypt depths were measured for each tissue section in this study.

### 2.7. Antioxidant Ability

The contents of superoxide dismutase (SOD), glutathione peroxidase (GSH-Px), total antioxidant capacity (T-AOC), and malondialdehyde (MDA) in the serum and jejunum were detected using kits (Kits A001-3, A015-2, A005-1, A003-1, respectively; Nanjing Jiancheng Bioengineering Institute, Nanjing, China). The methods entailed the use of the xanthine-oxidase–xanthine reaction, reduced glutathione, ammonium molybdate, and 2-thiobarbituric acid for the determination of SOD, GSH-Px, CAT, and MDA, respectively.

### 2.8. Quantitative Real-Time PCR

Total RNA was extracted from the jejunal mucosa using total RNA extraction reagent (Vazyme, Nanjing, China). We used agarose gel electrophoresis and ultramicro spectrophotometry (NanoDrop 2000, Thermo Fisher Scientific, Wilmington, NC, USA), respectively, to assess the quality of RNA samples and concentration. Then, we used 1 μg in the reaction system of the RNA samples in 20 μL cDNA synthesis. The GAPDH and target gene expressions were detected with a CFX connect real-time PCR detection system (CFX-96, Bio-Rad, Hercules, CA, USA) using SYBR green mixture (Q711-02, Vazyme, Nanjing, China). The primers of the target gene were designed using NCBI/Primer-BLAST online platform (https://www.ncbi.nlm.nih.gov/tools/primer-blast/index.cgi?LINK_LOC=BlastHome (accessed on 20 April 2023)), and the primer specificity was verified. The primers are listed in Table 2. Each sample hole add 10 μL reaction mixture, including 5 μL of SYBR green mixture, 0.3 μL each of the forward and reverse primers, 2 μL of diluted cDNA template, and 2.4 μL of nuclease-free water. Three technical replicates were conducted for each sample. The PCR protocol used was as follows: a first step of predenaturation at 95 °C for 30 s, a second step of denaturation at 95 °C for 5 s, and annealing at 60 °C for 30 S, and 40 cycles, and a third step of extension at 72 °C for 60 s. The correlation coefficient of the target gene amplification curve is required to be no less than 99%, and the amplification efficiency is between 90% and 110%. Analysis of the melting curve at the end of the amplification ensured that only one amplification product was synthesized. GAPDH was chosen as the reference gene to normalize the target gene and, accordingly, to calculate the relative target gene expression via the 2^−ΔΔCT^ method.

### 2.9. Statistical Analysis

All acquired data are presented as means. The effect of NCG on the variables was evaluated using independent sample *t*-tests. SPSS version 21.0 (SPSS Inc., Chicago, IL, USA) was used for the statistical analysis. A significance level of *p* < 0.05 was deemed indicative of a significant difference.

## 3. Results

### 3.1. Effects of NCG on Growth Performance in Weaned Pigs

Table 3 shows the effect of dietary NCG on the growth performance of weaned pigs. Compared to the control group, the NCG treatment group exhibited elevated ADG during the 0–14-day period, as well as improved ADG and average weight gain (AWG) during the 0–28-day interval (*p* < 0.05).

### 3.2. Effects of NCG on Apparent Nutrient Digestibility in Weaned Pigs

Table 4 presents the effects of dietary NCG supplementation on the apparent nutrient digestibility of weaned pigs. In contrast to the control group, the NCG treatment group displayed significant increases in DM and CP digestibility from day 15 to day 28 (*p* < 0.05).

### 3.3. Effects of NCG on Serum Biochemical Parameters in Weaned Pigs

Table 5 outlines the effects of dietary NCG supplementation on serum biochemical parameters in weaned pigs. The experimental group exhibited noteworthy enhancements in serum levels of GLU and HDL, along with reduced SUN levels, compared to the control group (*p* < 0.05).

### 3.4. Effects of NCG on Jejunal Morphology in Weaned Pigs

Table 6 shows the effects of dietary NCG supplementation on the jejunal morphology of weaned pigs. In comparison with the control group, NCG treatment led to a significant increase in VH, VH:CD ratio and jejunal wall thickness (*p* < 0.05).

### 3.5. Effects of NCG on Tight Junction Protein in Weaned Pigs

Figure 1 illustrates the modulation of tight junction protein mRNA expression in the jejunum via NCG intervention. The relative mRNA expression of jejunal *claudin-1*, *occludin*, and *ZO-1* was markedly increased in the NCG treatment group compared to that in the control group (*p* < 0.05).

### 3.6. Effects of NCG on Jejunal Amino Acid Transporters in Weaned Pigs

Figure 2 shows the alterations in jejunal amino acid transporter expression following dietary NCG supplementation. In contrast to the control group, the NCG group showed a substantial increase in the relative mRNA expression of *SNAT2*, *EAAC1*, *SLC3A1*, and *SLC3A2* (*p* < 0.05).

### 3.7. Effects of NCG on Antioxidant Ability of Serum and Jejunum in Weaned Pigs

Table 7 shows the effect of NCG on the antioxidant capacity of the serum and jejunum of weaned pigs. The NCG treatment group displayed significantly augmented levels of GSH-Px in the serum, accompanied by a notable reduction in MDA content (*p* < 0.05). The NCG treatment group displayed significantly augmented levels of T-AOC and SOD in the jejunum, accompanied by a notable reduction in MDA content (*p* < 0.05).

### 3.8. Effect of NCG on mRNA Expression of Antioxidant-Related Genes in the Jejunum of Weaned Pigs

Figure 3 shows the alteration in the mRNA expression of antioxidant-related genes within the jejunum after NCG intervention. The NCG treatment group exhibited an upregulation of the mRNA abundance of *Nrf2*, *AhR*, and *CYP1A1*, along with an increase in the mRNA expression of *CAT*, *SOD1*, *Gpx4*, *GCLC*, *GCLM*, and *NQO1* (*p* < 0.05), compared to the control group.

### 3.9. Effects of NCG on mRNA Expression of Jejunal Mucosa Cytokines in Weaned Pigs

Figure 4 shows the effect of NCG on cytokine mRNA expression in the jejunal mucosa of the weaned pigs. The NCG treatment group demonstrated a downregulation of *IL-1β* and *IL-4* mRNA expression, coupled with an upregulation of *IL-10* mRNA expression in the jejunum, compared to the control group (*p* > 0.05).

## 4. Discussion

The weaning phase is a crucial period that significantly affects the growth and development of piglets. The abrupt shifts in diet and living conditions during this transition can readily lead to piglet intestinal dyspepsia and related issues, profoundly affecting the economic viability of large-scale production [16,17]. Notably, arginine is the first rate-limiting amino acid in piglets and exerts a pivotal influence on intestinal growth, development, and nutrient absorption [18]. It has been observed that the activity of NAG stands as the primary limiting factor impeding endogenous arginine synthesis in piglets [19]. In this context, NCG effectively supplemented NAG within the urea cycle, thereby increasing endogenous arginine synthesis. Consequently, the role of NCG in ameliorating gastrointestinal health in young animals has garnered considerable attention in the industry. In the current study, the dietary incorporation of NCG yielded discernible improvements, elevating ADG during the 0–14 day span and showcasing augmented ADG and AWG throughout the 0–28 day timeframe post weaning. Moreover, NCG supplementation effectively enhanced the digestibility of DM and CP during days 15–28 post weaning, underscoring its potential to improve intestinal nutrient digestion and absorption efficacy among piglets. The findings of this study align with those of Wu et al. [6]., affirming the positive effect of NCG on growth performance and nutrient utilization in post-weaning piglets.

Serum biochemical parameters are reflective indicators of metabolic shifts, tissue permeability alterations, and the dynamics of nutrient digestion and absorption within the intestines [20]. GLU, as the principal energy contributor, assumes a multifaceted role in intestinal cell proliferation and adaptation, thereby exerting far-reaching effects on piglet intestinal development and nutrient processing [21]. HDL primarily functions in transporting cholesterol from the peripheral tissues to the liver for metabolic processing, ultimately contributing to the regulation of the serum total cholesterol levels [22]. SUN represents an end-product of protein or amino acid metabolism, and its diminished concentration indicates a heightened protein utilization from dietary sources and an increased nitrogen deposition within the organism [23]. Notably, the present study revealed that NCG supplementation triggered an elevation in serum GLU and HDL levels, accompanied by a significant reduction in SUN levels. These findings strongly indicate that NCG fosters enhanced GLU transport via small-intestinal epithelial cells, propelling amino acid absorption within the organism, while concurrently fortifying energy metabolism.

Morphological analysis serves as a pivotal indicator for evaluating jejunum growth and development, as well as the post-weaning digestion and absorption capability of piglets [24]. Changes in the intestinal villus height, crypt depth, and intestinal wall thickness are key markers of nutrient digestion and absorption efficiency. Moreover, augmenting the intestinal wall thickness can mitigate susceptibility to intestinal ailments [1,24,25]. In this study, dietary supplementation with NCG demonstrated the potential to enhance villus height, intestinal wall thickness, and VH:CD ratio in the jejunum of weanling piglets.

Tight junction proteins, including occludin, claudin-1, ZO-1, and other proteins, form essential constituents of the intestinal mucosal mechanical barrier. They play a pivotal role in upholding the jejunal mucosal permeability [26]. Claudin notably governs cell junctions and adhesion, whereas occludin integration into tight junctions reduces associated membrane permeability, thus safeguarding the intestinal mucosal barrier [27]. ZO-1 plays a crucial role in both maintaining and regulating the integrity of the intestinal epithelial barrier and its barrier function, and participates in important processes, such as cell material transport [28]. In the present study, it was observed that NCG supplementation led to an upregulation of mRNA expression for *occludin*, *claudin-1*, and *ZO-1*. Taken together with the jejunal amino acid transporters and morphological observations, this suggests that NCG may facilitate the absorption of nutrients within the jejunal mucosa and maintain the integrity of the jejunal mucosa. Zhang et al. [28]. conducted a study that reported the potential of NCG to enhance the integrity of the lamb jejunum, with similar results to those of this study.

The jejunum is the principal site for the absorption of most nutrients, including amino acids, into the bloodstream. Amino acid transporters serve as conduits for amino acid movement and orchestrate intracellular metabolic processes by sensing fluctuations in extracellular amino acids [29,30]. Notably, SNAT2 plays a pivotal role in regulating intracellular concentrations of anabolic amino acids [31]. EAAC1, a sodium-dependent high-affinity glutamate transporter, plays a central role in preserving the glutamate equilibrium. Solute carrier family 3 member 1 (SLC3A1) and solute carrier family 3 member 2 (SLC3A2) influence arginine absorption [32]. Remarkably, NCG supplementation resulted in the upregulation of the mRNA expression of *SNAT2*, *EAAC1*, *SLC3A1*, and *SLC3A2* in the jejunum. This augmentation signifies the potential of NCG to elevate the amino acid transport efficiency within the jejunum. The increased expression of *EAAC1* mRNA may be due to the regulatory effect of NCG on glutamate-to-arginine metabolism through the urea cycle, thereby intensifying the requirement for glutamate and augmenting its transport within the jejunum. This resemblance to the outcomes reported by Yang et al. [33]. suggests that NCG could potentially foster nutrient absorption by modulating the expression of amino acid transporters within the jejunum.

Subsequent investigations revealed that dietary NCG supplementation resulted in increased serum GSH-Px levels and diminished serum MDA levels. The intestinal mucosal structure and function can be susceptible to damage, which, in turn, affects both the intestinal development and overall health. The enzymatic system, comprising SOD, GSH-Px, and the total antioxidant capacity, plays a crucial role in counteracting oxygen free radicals. The accumulation of MDA in an organism indirectly mirrors the extent of oxidative damage in tissues, with elevated MDA levels indicating more pronounced damage [34]. The present study revealed that dietary supplementation with 500 mg/kg NCG augmented SOD and GSH-Px activity in both the serum and jejunal mucosa, while reducing MDA levels. A further examination of antioxidant enzyme-related gene expression in the jejunum indicated an upregulation of the mRNA expression of antioxidant enzymes (*CAT*, *SOD2*, *GPX4*, *GCLC*, and *GCLM*) upon NCG addition. This underscores NCG’s potential to enhance antioxidant enzyme activity and improve organismal antioxidant capacities.

Furthermore, considering the enhancement of the jejunal mucosa antioxidant capacity by NCG, we investigated the expression levels of antioxidation-related regulatory genes to elucidate the potential mechanisms through which NCG promotes the antioxidant system. Nrf2 is a pivotal transcription factor responsible for orchestrating the balance within the organism’s redox system. When oxidative stress stimulates an organism, Nrf2 expression escalates and dissociates from Keap1. This leads to the translocation of Nrf2 to the nucleus, where it forms a dimer with a small Maf protein, thereby activating the transcription of antioxidant enzyme genes [35]. AhR, a ligand-dependent transcription factor, oversees the transcription of CYP1A1, a catalyst for the production of reactive oxygen species, thereby increasing Nrf2 levels [36]. In the present study, it was revealed that NCG increased the mRNA expression of *AhR*, *CYP1A1*, and *Nrf2*. However, there was no significant impact on the mRNA expression of Keap1. This observation suggests that NCG potentially increases *CYP1A1* expression and reinforces the organism’s response to oxidative stress through AhR activation. This could lead to an upregulation of the expression of *Nrf2*, facilitating its translocation to the nucleus, and amplifying the expression of genes related to antioxidant enzymes.

Nrf2 can not only regulate the body’s REDOX level, but also regulate the expression of related immune factors in the jejunum, which may be due to Nrf2 alleviating the intestinal damage caused by weaning stress [37,38]. Among these immune factors, TNF-α, IL-1β, and IL-6 play prominent roles in inflammatory responses by stimulating cytokine release from intestinal immune cells. In contrast, IL-10, a vital anti-inflammatory factor within the organism, exerts both immunomodulatory and anti-inflammatory effects [39]. Within the scope of this study, it was demonstrated that dietary NCG supplementation upregulated the anti-inflammatory cytokine *IL-10*, and simultaneously downregulated the mRNA expression of *IL-1β* and *IL-6* in weaned pigs. This anti-inflammatory effect of NCG could be achieved through the upregulation of *Nrf2* expression, which subsequently influences immune factors and contributes to the amelioration of weaning-induced inflammation.

## 5. Conclusions

In summary, the present study established that the inclusion of dietary NCG at a dose of 500 mg/kg yielded a range of beneficial outcomes. These include an increase in ADG, an enhanced digestibility of DM and CP, and elevated serum GLU and HDL levels. Furthermore, NCG supplementation demonstrates the capacity to enhance the mRNA expression of jejunal amino acid transporters, consequently promoting an improved jejunal morphology and improved antioxidant properties in weaned pigs. Moreover, the findings might indicate that NCG contributes to an increase in Nrf2 mRNA expression. This potentially underpins the enhancement of both the antioxidant capacity and the expression of inflammatory cytokines within the jejunum mucosa.

## Figures and Tables

**Figure 1 animals-13-03183-f001:**
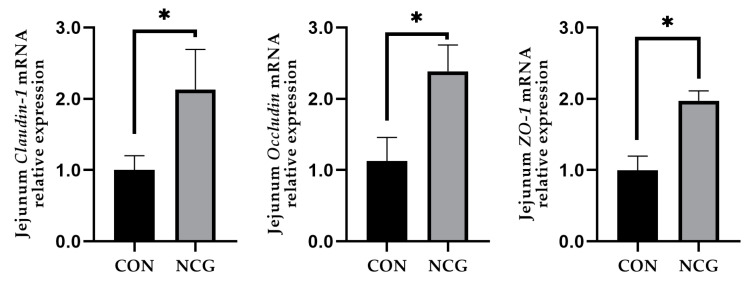
The effects of NCG on jejunal tight junction protein mRNA expression in weaned pigs. CON = control treatment, NCG = 500 mg/kg NCG added to the diet on the basis of the control treatment. Values are expressed as means ± SEM, *n* = 6, * *p* < 0.05. *ZO-1*, tight junction proteins 1.

**Figure 2 animals-13-03183-f002:**
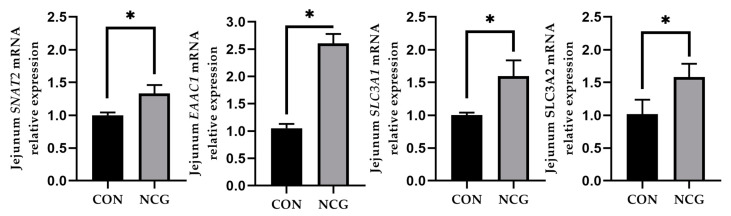
The effects of NCG on jejunal amino acid transporters in weaned pigs. CON = control treatment, NCG = 500 mg/kg NCG added to the diet on the basis of control treatment. Values are expressed as means ± SEM, *n* = 6, * *p* < 0.05. *SNAT2*, solute carrier family 38 member 2; *EAAC1*, solute carrier family 1 member 1; *SLC3A1*, solute carrier family 3 member 1; *SLC3A2*, solute carrier family 3 member 2.

**Figure 3 animals-13-03183-f003:**
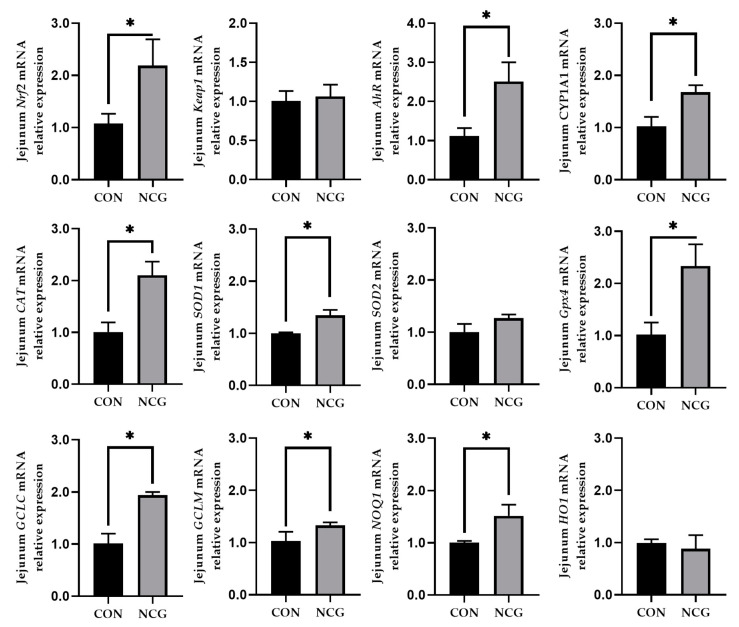
The effect of NCG on the mRNA expression of antioxidant genes in the jejunum of weaned pigs. CON = control treatment, NCG = 500 mg/kg NCG added to the diet on the basis of the control treatment. Values are expressed as means ± SEM, *n* = 6, * *p* < 0.05. *SOD1*, superoxide dismutase 1; *SOD2*, superoxide dismutase 2; *CAT*, catalase; *Gpx4*, glutathione peroxidase 4; *Nrf2*, nuclear factor erythroid 2-related factor 2; *Keap1*, Kelch-like ECH-associated protein 1; *AhR*, aryl hydrocarbon receptor; *CYP1A1*, cytochrome P450 family 1 subfamily A member 1; *NQO1*, NAD(P)H quinone dehydrogenase 1; *GCLC*, glutamate cysteine ligase catalytic subunit; *GCLM*, glutamate cysteine ligase modifier subunit; *HO-1*, heme oxygenase 1.

**Figure 4 animals-13-03183-f004:**
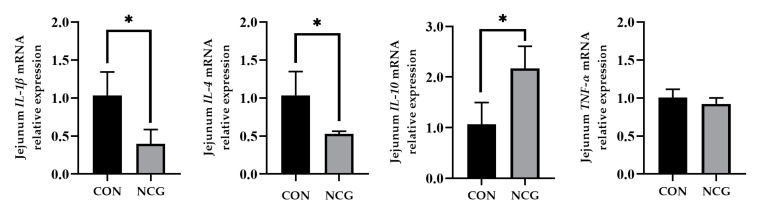
The effects of NCG on the mRNA expression of jejunal mucosa cytokines in weaned pigs. CON = control treatment, NCG = 500 mg/kg NCG added to the diet on the basis of the control treatment. Values are expressed as means ± SEM, *n* = 6, * *p* < 0.05. *TNF-α*, tumour necrosis factor α; *IL-1β*, interleukin 1β; *IL-6*, interleukin 6; *IL-10*, interleukin 10.

**Table 1 animals-13-03183-t001:** Composition and nutrient levels of the basal diet (air-dry basis) (%).

Ingredients	Content	Nutrient Levels	Content
Corn	60.07	Digestible energy (MJ/kg) ^2^	14.32
Soybean meal	15.00	CP	17.93
Wheat bran	4.00	Ca	0.65
Extruded full-fat soybean	9.00	P	0.50
Concentrated soybean protein	3.00	Lys	1.28
Whey powder	3.00	Met	0.42
Soybean oil	1.00	Thr	0.84
Sucrose	1.50	Trp	0.25
Limestone	0.60		
CaHPO_4_	1.20		
L-Lysiine HCl	0.45		
DL-Methionine	0.15		
L-Threonine	0.15		
L-Tryptophan	0.05		
L-Valine	0.03		
NaCl	0.30		
Vitamin–mineral premix ^1^	0.50		
Total	100.00		

^1^ Provided per kilogram of diet: VA 2700 IU, VD_3_ 300 IU, VE 25 IU, VK_3_ 0.5 mg, VB_1_ 1.50 mg, VB_2_ 5.25 mg, VB_12_ 0.026 mg, folic acid 0.3 mg, nicotinic 22 mg, pantothenate 14 mg, biotin 0.05 mg, Cu 5.6 mg, Fe 90 mg, Mn 5.2 mg, Zn 106 mg, Se 0.3 mg, I 0.2 mg. ^2^ The digestible energy was the calculated value, and the other nutrient levels were the analyzed value.

**Table 2 animals-13-03183-t002:** Primers used for quantitative reverse transcription PCR ^1^.

Gene ^2^	Primer Sequences (5′→3′)	Product Size (bp)	Accession Number
*GAPDH*	F:GAAGGTCGGAGTGAACGGAT R:CATGGGTAGAATCATACTGGAACA	149	AF017079
*SOD1*	F:AAGGCCGTGTGTGTGCTGAA R:GATCACCTTCAGCCAGTCCTTT	118	NM_001190422.1
*SOD2*	F:GGCCTACGTGAACAACCTGA R:TGATTGATGTGGCCTCCACC	126	NM_214127.2
*CAT*	F:ATCCAGCCAGTGACCAGATG R:CCCGGTCAAAGTGAGCCATT	183	NM_214301.2
*Gpx4*	F:TGATAAGAACGGCTGTGTGGT R:TTGAGCTAGAGGTAGCACGG	90	NM_214407.1
*Nrf2*	F:TGCTTTATAGCGTGCAAACCT R:TGTCAATCAAATCCATGTCCCTTG	191	XM_021075133.1
*Keap1*	F:CGTGGAGACAGAAACGTGGA R:CAATCTGCTTCCGACAGGGT	239	XM_021076667.1
*AhR*	F:TTGCTACAGGCTCTAAATGGCTT	209	NM_001303026.1
	R: GAGTCTGGACACTGTGAAGGG		
*CYP1A1*	F:CCTTCACCATCCCTCACAGT	196	NM_214412.1
	R:ATCACCTTTTCACCCAGTGC		
*NQO1*	F:CATGGCGGTCAGAAAAGCAC R:ATGGCATACAGGTCCGACAC	135	NM_001159613.1
*GCLC*	F:CTCCCTTGTGGTACCTCTGC	255	XM_021098556.1
	R:GTCCTCCACCGTGTTGAACT		
*GCLM*	F:AGTTCACCGTCCTGCCAAAT	189	XM_001926378.4
	R:AACAGACATAGCCTGCCACC		
*HO-1*	F:TACCGCTCCCGAATGAACAC	209	NM_001004027.1
	R:GTCACGGGAGTGGAGTCTTG		
*Claudin-1*	F:CTCTCCCCACATTCGAGATGATT R:AGCCATTGACGTGATCCAGG	247	NM_001244539.1
*Occludin*	F:TCAGGTGCACCCTCCAGATT R:TATGTCGTTGCTGGGTGCAT	169	NM_001163647.2
*ZO-1*	F:CCAACCATGTCTTGAAGCAGC R:TGCAGGAGTGTGGTCTTCAC	215	XM_021098896.1
*TNF-α*	F:GCCCTTCCACCAACGTTTTC R:CAAGGGCTCTTGATGGCAGA	97	NM_214022.1
*IL-1β*	F:ACACACCTCTGACTCAAGCC R:GGGGCCATCAGCCTCAAATA	275	NM_001302388.2
*IL-6*	F:TTCAGTCCAGTCGCCTTCT	91	NM_214399.1
	R:GTGGCATCACCTTTGGCATCTTCTT		
*IL-10*	F:CACTGCTCTATTGCCTGATCTTCC	136	NM_214041.1
	R:AAACTCTTCACTGGGCCGAAG		
*SNAT2*	F:TGATTCTTGCCGACCACGTT	128	XM_003126626.6
	R:GGCTGCGGCCTTTAAGTTCT		
*EAAC1*	F:TTGGTCTACGTGCTGTCGTATATT	263	NM_001164649.1
	R:CGTCTCTGGCTCACTAGAAGG		
*SLC3A1*	F:TACCCACCCGGTCAGACTAC	188	NM_001123042.1
	R:TATGCCGTTGAGCTCTCTCG		
*SLC3A2*	F:GACCCCGCTTTCGGTTCTAA	294	XM_003353809.4
	R:AATCAAGAGCCTATCCTCGCTG		

^1^ PCR, polymerase chain reaction. ^2^
*GAPDH*, glyceraldehyde-3-phosphate dehydrogenase; *SOD1*, superoxide dismutase 1; *SOD2*, superoxide dismutase 2; *CAT*, catalase; *Gpx4*, glutathione peroxidase 4; *Nrf2*, nuclear factor erythroid 2-related factor 2; *Keap1*, Kelch-like ECH-associated protein 1; *AhR*, aryl hydrocarbon receptor; *CYP1A1*, cytochrome P450 family 1 subfamily A member 1; *NQO1*, NAD(P)H quinone dehydrogenase 1; *GCLC*, glutamate cysteine ligase catalytic subunit; *GCLM*, glutamate cysteine ligase modifier subunit; *HO-1*, heme oxygenase 1; *TNF-α*, tumour necrosis factor α; *IL-1β*, interleukin 1β; *IL-6*, interleukin 6; *IL-10*, interleukin 10; *SNAT2*, solute carrier family 38 member 2; *EAAC1*, solute carrier family 1 member 1; *SLC3A1*, solute carrier family 3 member 1; *SLC3A2*, solute carrier family 3 member 2.

**Table 3 animals-13-03183-t003:** The effects of dietary NCG on the growth performance of weaned pigs.

Items	Treatment ^1^		SEM	*p*-Value ^2^
	CON	NCG		
Initial BW, kg	5.69	5.71	0.17	0.87
Final BW, kg	13.14	13.91	0.49	0.13
0–14 d				
ADFI, g	360.18	394.60	30.63	0.12
ADG, g	210.32 ^b^	235.32 ^a^	11.38	0.03
F:G ratio	1.71	1.68	0.06	0.66
15–28 d				
ADFI, g	550.38	583.64	30.77	0.29
ADG, g	322.22	349.84	17.43	0.12
F:G ratio	1.77	1.67	0.11	0.41
0–28 d				
AWG/kg	7.46 ^b^	8.19 ^a^	0.35	0.04
ADFI, g	455.28	489.12	20.50	0.11
ADG, g	266.27 ^b^	292.58 ^a^	12.61	0.04
F:G ratio	1.74	1.68	0.08	0.42

^1^ CON = control treatment, NCG = 500 mg/kg NCG added to the diet on the basis of control treatment. ^2^ Mean values within a same row between ^a^ and ^b^ were significantly different when *p* < 0.05, *n* = 6. BW, body weight; ADG, average daily gain; ADFI, average daily feed intake; AWG, average weight gain; F:G ratio, ADFI: ADG.

**Table 4 animals-13-03183-t004:** The effects of dietary NCG on the apparent nutrient digestibility of weaned pigs.

Items	Treatment ^1^		SEM	*p*-Value ^2^
	CON	NCG		
0–14 d				
DM, %	84.78	85.58	1.22	0.52
CP, %	77.07	80.04	1.60	0.07
EE, %	71.46	71.73	4.44	0.95
NDF, %	50.27	53.85	2.79	0.21
ADF, %	24.40	29.15	2.85	0.10
Ca, %	49.23	50.06	1.99	0.68
P, %	30.18	31.00	1.61	0.61
15–28 d				
DM, %	86.35 ^b^	89.52 ^a^	0.64	0.01
CP, %	84.49 ^b^	87.23 ^a^	0.37	0.01
EE, %	74.15	74.52	3.89	0.92
NDF, %	51.94	55.16	2.56	0.22
ADF, %	28.74	31.37	4.24	0.54
Ca, %	51.24	53.10	1.50	0.22
P, %	31.75	31.53	1.07	0.84

^1^ CON = control treatment, NCG = 500 mg/kg NCG added to the diet on the basis of the control treatment. ^2^ Mean values within a same row between ^a^ and ^b^ were significantly different when *p* < 0.05, *n* = 6. DM, dry matter; CP, crude protein; EE, ether extract; NDF, neutral detergent fiber; ADF, acid detergent fiber; Ca, calcium; P, phosphorus.

**Table 5 animals-13-03183-t005:** The effects of NCG on the serum biochemical parameters in weaned pigs.

Items	Treatment ^1^		SEM	*p*-Value ^2^
	CON	NCG		
GLU, mmol/L	7.24 ^b^	8.11 ^a^	0.27	0.01
TP, g/L	55.40	56.13	1.34	0.61
ALB, g/L	38.93	39.63	0.85	0.46
GLB, g/L	16.47	16.50	1.86	0.99
A:G, %	2.42	2.41	0.29	0.96
HDL, mmol/L	0.90 ^b^	1.15 ^a^	0.06	0.01
LDL, mmol/L	1.40	1.27	0.18	0.50
TC, mmol/L	2.21	2.26	0.33	0.89
TGs, mmol/L	0.92	0.71	0.09	0.13
SUN, mmol/L	3.90 ^a^	3.00 ^b^	0.28	0.03

^1^ CON = control treatment, NCG = 500 mg/kg NCG added to the diet on the basis of the control treatment. ^2^ Mean values within a same row between ^a^ and ^b^ were significantly different when *p* < 0.05, *n* = 6. GLU, glucose; TP, total protein; ALB, albumin; GLB, globulin; A:G ratio, ALB: GLB; HDL, high-density lipoprotein; LDL, low-density lipoprotein; TC, total cholesterol; TGs, triglycerides; SUN, serum urea nitrogen.

**Table 6 animals-13-03183-t006:** The effects of NCG on the jejunal morphology in weaned pigs.

Items	Treatment ^1^		SEM	*p*-Value ^2^
	CON	NCG		
VH, µm	313.83 ^b^	412.42 ^a^	16.30	0.01
CD, µm	282.62	273.40	9.04	0.33
VH/CD ratio	1.11 ^b^	1.51 ^a^	0.06	0.01
IWT, µm	141.32 ^b^	184.45 ^a^	9.70	0.01

^1^ CON = control treatment, NCG = 500 mg/kg NCG added to the diet on the basis of the control treatment. ^2^ Mean values within a same row between ^a^ and ^b^ were significantly different when *p* < 0.05, *n* = 6.

**Table 7 animals-13-03183-t007:** The effects of NCG on the antioxidant ability of the serum and jejunum in weaned pigs.

Items	Treatment ^1^		SEM	*p*-Value ^2^
	CON	NCG		
Serum				
T-AOC, μmol/g	159.78	172.97	9.75	0.29
SOD, U/mg prot	287.47	292.07	3.51	0.26
GSH-Px, U/mg	253.19 ^b^	305.06 ^a^	14.15	0.02
MDA, nmol/mg prot	5.00 ^a^	4.26 ^b^	0.19	0.02
Jejunum				
T-AOC, μmol/g	267.93 ^b^	339.98 ^a^	18.95	0.02
SOD, U/mg prot	354.39 ^b^	392.17 ^a^	6.32	0.01
GSH-Px, U/mg	9.77	10.57	0.91	0.43
MDA, nmol/mg prot	0.44 ^a^	0.29 ^b^	0.03	0.01

^1^ CON = control treatment, NCG = 500 mg/kg NCG added to the diet on the basis of the control treatment. ^2^ Mean values within a same row between ^a^ and ^b^ were significantly different when *p* < 0.05, *n* = 6. T-AOC, total antioxidant capacity; SOD, superoxide dismutase; GSH-Px, glutathione peroxidase; MDA, malondialdehyde.

## Data Availability

The datasets analyzed are not publicly available due to ownership by the funding partners, but are available from the corresponding author on reasonable request.

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
