# Peer review of "Effect of N-Carbamylglutamate Supplementation on Growth Performance, Jejunal Morphology, Amino Acid Transporters, and Antioxidant Ability of Weaned Pigs"

_animals, 2023, doi:10.3390/ani13203183_

Round 1
Reviewer 1 Report
The present study aimed to investigate the impact of N-carbamylglutamate (NCG) on the growth performance and antioxidant capacity of weaned pigs. The study was conducted on 36 weaned pigs, which were randomly assigned to two treatment groups: control group and 500 mg/kg NCG group. The study suggests that NCG plays an important role in improving the growth performance, jejunal morphology, jejunal transport of amino acids related to the ornithine cycle, and antioxidant capacity of weaned pigs. Overall, the subject itself is surely worthy of investigation. However, there are several concerns that need to be addressed as follows:
- Throughout the manuscript, the writing style should be formal from the third-person perspective. Do not use “we” (e.g. in lines 21, 63, 335, … etc) or “our” (e.g. in lines 335, 340, 349… etc ).
- There is a problem in using the abbreviations throughout the manuscript, especially in the abstract. The abbreviation must be introduced upon the first mention of the full term followed by its abbreviation in parentheses: From then on, the abbreviation must be used exclusively and throughout.
- Simple summary section lacks important information regarding the overall results and conclusions of the study and how they will be valuable to society.
- Lines 20-21: Place the question addressed in a broad context and highlight the purpose of the study.
- Line 74; how was the NCG dose added to the control diet? Were the diets pelleted or mashed?
- The authors had to justify on what basis they selected the dietary level of NCG.
- Table 1: the sum of ingredients is not equal to 100%. Please revise. Also, remove “(%)” from Nutrient levels as it is already mentioned in the Table title.
- Line 93; Describe the piglet pens.
- Line 101; It is highly recommended to mention how the fecal samples were collected and how many samples were taken per treatment.
- Line 105: what is meant by “Upon culmination”?
- Line 108-110: Have piglets been slaughtered and what is the number of collected samples per treatment? Please clarify.
- I have concerns about the use of acid-insoluble ash (AIA) as an internal indicator to determine nutrient apparent digestibility in the study. The very low concentration of AIA in the basal diet's ingredients may lead to inaccurate results, as even small contents in its content can significantly impact the findings.
- Line 127; Specify If you added α amylase or not, and was their content expressed as exclusive or inclusive of residual ash?
- Lines 139- 141; The instruments and kits used must contain all full information such as model, company, city, country.
In all Figs., describe the experimental groups and all abbreviations used in the figure legends. Describe also the number of analyzed samples (n=?).
-
Reviewer 2 Report
The manuscript entitled “Effect of N-carbamylglutamate Supplementation on Growth Performance, Apparent Nutrient Digestibility and Jejunal Morphology, Amino Acid Transporters and Antioxidant Ability of 4 Weaned Pigs” by Naizhi Hu et al. investigated the effects of dietary supplementedN-carbamylglutamate on growth performance, apparent nutrient digestibility and jejunal morphology, amino acid transporters and antioxidant ability of weanling Pigs. The results showed that that dietary N-carbamylglutamate supplementation improved jejunal morphology and bolstered antioxidant properties in weaned pigs. The experiment is fit for the scope of Animals.The content of this manuscript was important for the pig production. The writing of the manuscript was very good. But I think the current manuscript needs to be clearly revised due to the following concerns.
1.Line 35: “antioxidant” → “antioxidant capacity”.
2. Line 35: antioxidant → antioxidant capacity.
3. The subheadings of lines 230 and 238 are repeated, so it is recommended to combine them for analysis.
4. It is suggested to list the calculation formulas of ADFI and ADG after line 100.
5. Line 101, details how many times a day to collect feces. At the same time, after drying the feces, whether there is moisture return?
6. Line 207, “Effects of dietary NCG on nutrient digestibility of weaned pig” →“Effects of dietary NCG on apparent nutrient digestibility of weaned pig”
7. There was a punctuation error on line 238
Minor editing of English language required
Reviewer 3 Report
The authors investigated the effects of N-carbamylglutamate (NCG) on the growth performance, apparent nutrient digestibility, jejunal morphology, amino acid transporters, and antioxidant capacity in weaned piglets. These findings provide valuable insights into the application of NCG in swine production. I have some minor suggestions for modification.
1. Line 2-4: The title of this manuscript is long, so suggested to delete “apparent nutrient digestibility”;
2. Line 56: Add Spaces before “(CPS-1)”
3. Line 59-61: The physiological structure of ruminants is quite different from that of monogastric animals. It is suggested to delete this descriptive language and the corresponding reference;
4. Line 91-92: The superscript of the table and figure should be consistent. Such as,“a” changed to“1”;
5. Line 121-122: The weight of fresh feces collected daily by using indicator method needed more details.
6. Line 125: The symbol is used incorrectly. For example, the symbol for temperature should be ℃, the same below;
7. Line 126: Which standard is referred to for the determination of crude protein content by Kjeldahl nitrogen determination method;
8. Line 137: Is it necessary to enumerate the formula for determining nutrient digestibility using the endogenous indicator?
9. Line 157-162: The indicators determined by the kit are not marked by the instrument used, please mark them in detail;
10. Line 327-333: I think this paragraph should be placed before the languages of jejunal amino acid transporters;
11. Line 379-389: The authors hypothesized that NCG's enhancement of Nrf2 expression might influence the expression of immune factors of pigs. What could be the mechanism?
no comments
Reviewer 4 Report
Dear Authors,
Congratulations for your work but please, I wonder if you clarify some points before going further with the review.
I am quite concerned with the design because the fact that animals were fed ad libitum produced important differences in feed intake since the first part of the trial and could have been enough to lead to those differences in performance and glucids metabolism. It can simply be explained by any palatability changed produced by the diet with NCG. Do you have any data about that or Please, could you present the data of feed intake on the initial days of the study, specially first and second, to verify the NCG diet did not increase intake because of is taste and just because any other thing.
In the study you do not explain how pigs are housed. Individual, groups. Could you explain that? Are feed intake values group figures or individual ones.
Can you explain why animals were fed ad-lib instead of restricted. The higher feeding level on the NCG treatment is reason enough to justify a different FCR, ADG and glucid metabolism. Then, some of these changes, are they product of the NCG or the significantly different level of feed intake? Are changes in digestibility an effect of that?
Please, could you state which were the age of the pigs at the beggining of the trial and the average and SE of each group? Could you clarify if the animals were fed with the basal diet prior to the study? Do you have information about their feed intake prior to the challenge?
LINE 80. It says the two treatments were allocated based in weight; please can you further describe how it was and why it wasn’t a bias for the study? Perhaps they were assigned based in weight to balance both treatments – opposite to random assignation – but then you should remake the sentence cause it is not clear as currently written.
LINE 80: please state if the animals were housed alone or in groups. Please state the age of the pigs at the beginning on the trial and if the animals allocated to the different groups had similar age as there are important differences in gut maturity that could affect how nutrients are digested.
TABLE 3: SEM on 0-14 d ADFI must be wrong, it may be closer to 30 instead 306.
LINE 282- Not compulsory, though I recommend to change the word garnered as it is rare and it use and knowledge is very limited for not native speakers. “Obtained” is much more common, “got” is the simplest. Similar issue with “bolster” in line 287, though this one is not as mush rare but still not frequent.
Line 289; Nutrient utilization is not proved unless you explain how digestibility was correctly measured. Actually, differences in digestibility reported only affect DM and CP. DM and CP apparent digestibility can be affected by endogenous losses and different levels of feed intake– NCG ate more than CON so the endogenous produced protein should be more concentrated on CON digestive material -assuming a similar level of endogenous prot production-. There are not any other differences on the other nutrients so it is very questionable that the statement that the results align with the fact that NCG achieved a better nutrient utilization in this trial. Pigs grew more cause they ate more, and the digestibility study does not produce enough significant differences to support that statement.
LINE 293-294. The statement about GLU is not something proved in this study so it should be sustained by a reference.
LINE 302. Pigs on the NCG treatment ate an average of 34 extra grammes per day (approx. a 8% more). That can have an important effect on the metabolism and use of nutrients including digestibility and levels of nutrients on blood. How can you explain that levels of GLU were a direct effect of the NCG treatment or were an effect that the NCG treatment also produced and increase on feed intake.
Expression; Please, could you explain how you determined it was an upregulation of the expression of the gen or the tissue just contained more cells expressing that gen at the same pace.
Round 2
Reviewer 1 Report
All comments have been addressed.
Reviewer 4 Report
Congrats for your work !!